# Radiomics Analysis of ^18^F-FDG PET/CT for Prognosis Prediction in Patients with Stage III Non-Small Cell Lung Cancer Undergoing Neoadjuvant Chemoradiation Therapy Followed by Surgery

**DOI:** 10.3390/cancers15072012

**Published:** 2023-03-28

**Authors:** Jang Yoo, Jaeho Lee, Miju Cheon, Hojoong Kim, Yong Soo Choi, Hongryull Pyo, Myung-Ju Ahn, Joon Young Choi

**Affiliations:** 1Department of Nuclear Medicine, Veterans Health Service Medical Center, Seoul 05368, Republic of Korea; 2Department of Preventive Medicine, Seoul National University College of Medicine, Seoul 03080, Republic of Korea; 3Division of Pulmonary and Critical Care Medicine, Department of Medicine, Samsung Medical Center, Sungkyunkwan University School of Medicine, Seoul 06351, Republic of Korea; 4Department of Thoracic and Cardiovascular Surgery, Samsung Medical Center, Sungkyunkwan University School of Medicine, Seoul 06351, Republic of Korea; 5Department of Radiation Oncology, Samsung Medical Center, Sungkyunkwan University School of Medicine, Seoul 06351, Republic of Korea; 6Division of Hematology-Oncology, Department of Medicine, Samsung Medical Center, Sungkyunkwan University School of Medicine, Seoul 06351, Republic of Korea; 7Department of Nuclear Medicine, Samsung Medical Center, Sungkyunkwan University School of Medicine, Seoul 06351, Republic of Korea

**Keywords:** non-small cell lung cancer, ^18^F-FDG PET/CT, radiomics, overall survival, LASSO score, decision curve analysis

## Abstract

**Simple Summary:**

The estimation of overall survival (OS) in patients with stage III non-small cell lung cancer (NSCLC) is very important for determining a precise therapeutic strategy. Radiomics is a promising method to extract features that can reflect distinct differences in tumor phenotype from image data. We evaluated the prognostic impact of the radiomic features from ^18^F-FDG PET/CT to predict OS in patients with stage III NSCLC undergoing neoadjuvant concurrent chemoradiation therapy (CCRT) followed by surgery and compared the predictive performance of radiomics versus conventional PET parameters. We demonstrated that radiomic features using ^18^F-FDG PET/CT could be robust and useful in assessing the survival rate. Furthermore, this study demonstrated the important prognostic implications of radiomics in ^18^F-FDG PET/CT after neoadjuvant CCRT as well as pretreatment examination. The newly developed LASSO score using radiomic features performs better for individualized OS estimation than conventional PET parameters.

**Abstract:**

We investigated the prognostic significance of radiomic features from ^18^F-FDG PET/CT to predict overall survival (OS) in patients with stage III NSCLC undergoing neoadjuvant chemoradiation therapy followed by surgery. We enrolled 300 patients with stage III NSCLC who underwent PET/CT at the initial work-up (PET1) and after neoadjuvant concurrent chemoradiotherapy (PET2). Radiomic primary tumor features were subjected to LASSO regression to select the most useful prognostic features of OS. The prognostic significance of the LASSO score and conventional PET parameters was assessed by Cox proportional hazards regression analysis. In conventional PET parameters, metabolic tumor volume (MTV) and total lesion glycolysis (TLG) of each PET1 and PET2 were significantly associated with OS. In addition, both the PET1-LASSO score and the PET2-LASSO score were significantly associated with OS. In multivariate Cox regression analysis, only the PET2-LASSO score was an independently significant factor for OS. The LASSO score showed better predictive performance for OS regarding the time-dependent receiver operating characteristic curve and decision curve analysis than conventional PET parameters. Radiomic features from PET/CT were an independent prognostic factor for the estimation of OS in stage III NSCLC. The newly developed LASSO score using radiomic features showed better prognostic results for individualized OS estimation than conventional PET parameters.

## 1. Introduction

Non-small cell lung cancer (NSCLC) is the leading cause of cancer-related death worldwide, in spite of major advances in treatment [1]. Approximately 30% of all NSCLC is stage III, with larger tumor size or metastatic lymph nodes in the mediastinum, and it represents a diverse range of disease, from those with potentially operable stages to those with unresectable advanced stages [2]. The overall prognosis of stage III NSCLC is still poor despite multimodal treatment, and the expected 5-year survival for stage III NSCLC ranges from 13% to 36% [3]. These days, neoadjuvant concurrent chemoradiotherapy (CCRT) followed by surgery has been attempted, and the overall outcome has improved by preventing the rate of local failures and distant metastasis [4,5,6,7]. 

As ^18^F-fluorodeoxyglucose positron emission tomography/computed tomography (^18^F-FDG PET/CT) imaging has established itself as an important modality for patients with NSCLC, metabolic parameters including not only the semi-quantitative an index, standardized uptake value (SUV), but also volume-based PET parameters such as metabolic tumor volume (MTV) and total lesion glycolysis (TLG), have been suggested to be meaningful prognostic factors in NSCLC [8,9,10]. However, these conventional PET parameters can only represent gross tumor glucose metabolism and do not represent the detailed diversity of metabolism in heterogeneous components of the tumor. Tumor shapes are extremely variable and asymmetrical, especially in the advanced stages of NSCLC. The intratumoral ^18^F-FDG uptake can also be highly heterogeneous due to mixed tumor cells comprised of aggressive and nonviable cells, such as necrosis and fibrotic scar, respectively. 

Radiomics is a promising method to extract features that can display distinct differences in tumor phenotype and determine subtle information on the tumor characteristics and microenvironment at the cellular level based on imaging data [11,12,13]. It has been reported that radiomics is significantly related to the clinical diagnosis of metastatic lymph nodes and treatment response in NSCLC [14,15]. To the best of our knowledge, however, there is no study to investigate the prognostic significance of the radiomic features in NSCLC. Therefore, we evaluated the prognostic impact of the radiomic features extracted from ^18^F-FDG PET/CT to predict overall survival (OS) in patients with stage III NSCLC undergoing neoadjuvant CCRT followed by surgery and compared the predictive performance of radiomics versus conventional PET parameters in this study. 

## 2. Materials and Methods

### 2.1. Subjects

We retrospectively examined the medical documents of all patients with newly diagnosed stage III NSCLC who underwent initial ^18^F-FDG PET/CT (PET1) for staging work-up between November 2008 and October 2020. They were staged according to the 8th edition of the TNM classification [16] and performed a second ^18^F-FDG PET/CT within approximately 3 weeks following the completion of neoadjuvant CCRT. The following inclusion criteria were applied: (1) pathologically confirmed primary NSCLC, (2) completion of an expected neoadjuvant CCRT and surgical treatment, (3) performance of a second ^18^F-FDG PET/CT (PET2) within 2 weeks prior to surgical treatment, and (4) treatment with adjuvant treatment (radiotherapy alone, chemotherapy alone, or combined chemoradiotherapy) after surgery. Patients who were previously diagnosed with another malignant disease were excluded. Patients who only underwent initial PET/CT or follow-up PET/CT alone were also excluded. The endpoint of this study was OS, calculated from the day of disease diagnosis to the date of death from any cause or the date of the last clinical follow-up. 

This retrospective study was approved by the institutional review board of Samsung Medical Center (IRB No. 2020-09-185), and a waiver of informed patient consent was obtained from the IRB. Clinical records and survival data were acquired from the patients’ medical documents and the institutional cancer registry database. 

### 2.2. ^18^F-FDG PET/CT Acquisition and Analysis

All patients were instructed to fast for at least six hours, and their blood glucose level was confirmed to be less than 200 mg/dL. Each patient was injected with an FDG dose of 5 MBq/kg. After the administration of FDG, patients were strictly instructed to rest for one hour before the scan. CT images were obtained first, and then PET images were obtained from the skull base to the thigh using a dedicated PET/CT scanner (Discovery STe, GE Healthcare, Waukesha, WI, USA). The CT images were performed using a 16-slice helical CT with the following protocol: 140 keV, 30~170 mAs with auto A-mode, and a slice thickness of 3.75 mm. PET images were acquired in 3D mode for a 2 min scan/bed position and were reconstructed with 3.0 mm slice thickness using an ordered-subset expectation-maximization algorithm (20 subsets and 2 iterations).

For the quantitative evaluation, the gradient-based segmentation for volumes of interest (VOIs) of primary lung tumors was performed using PET Edge in MIM version 6.4 (MIM Software Inc., Cleveland, OH, USA) [14,15]. We estimated maximum SUV (SUVmax), mean SUV (SUVmean), MTV, and TLG within the entire primary tumor as conventional PET parameters. The differences in these parameters between PET1 and PET2 were calculated by subtracting PET2 parameters from PET1 parameters and dividing by PET1 parameters. These VOISs were subsequently saved as a DICOM-RT that was integrated into the Chang-Gung Image Texture Analysis toolbox (CGITA, http://code.google.com/p/cgita (accessed on 10 March 2021), an open-source software package incorporated in MATLAB (version 2014b; MathWorks, Inc., Natick, MA, USA) to derive the radiomic features from both PET1 and PET2 images. Ten matrices were revealed in three dimensions, showing 72 radiomic features for each PET image (Appendix A).

Two board-certified nuclear medicine physicians (J.Y. and J.Y.C.) with more than 12 years of experience in reading PET/CT interpreted the neoadjuvant CCRT response regarding PERCIST 1.0 [17]. They were blinded to the other clinical information, including pathologic reports. 

### 2.3. Neoadjuvant CCRT and Histopathologic Findings

Neoadjuvant CCRT included chemotherapy and concurrent thoracic radiotherapy. The total dose of 44~45 Gy thoracic radiotherapy was applied to patients over 5 weeks. The chemotherapy protocols are mostly composed of intravenous injections of paclitaxel (50 mg/m^2^ per week) or docetaxel (20 mg/m^2^ per week) plus either cisplatin (25 mg/m^2^ per week) or carboplatin (AUC, 1.5 per week) for 5 weeks [15].

Surgery was scheduled within 4 to 6 weeks after the completion of neoadjuvant CCRT. The operation was performed mainly by resection of the affected lung with ipsilateral mediastinal/hilar lymph node dissection considering the clinical stage. Since then, the surgical samples were evaluated by oncologic pathologists for viable tumors. They recorded the percentage of residual tumor, assessed by comparing the viable tumor foci with the cross-sectional areas of necrosis, fibrosis, and inflammation, slice by slice [18,19]. 

### 2.4. Postoperative Treatment and Follow-Up

Postoperative radiotherapy and/or chemotherapy were optionally allowed in the following cases based on the pathologic results: metastatic mediastinal/hilar lymph nodes with extracapsular invasion, or close (less than 5 mm distance from stump to tumor) or positive resection margins. All patients were regularly checked up according to our follow-up protocol [4,7]. 

### 2.5. Feature Selection and Radiomic Feature Construction

We used the least absolute shrinkage and selection operator (LASSO) method using a 10-fold cross-validation for minimizing overfitting in order to select the most useful prognostic variables on the association between radiomic features and OS of the patients [20]. A LASSO score was calculated for each patient through a linear combination of selected features weighted by their respective coefficients. In the same way, each LASSO score for radiomic features extracted from PET1 and PET2 was obtained. 

### 2.6. Statistical Analysis

All statistical methods were assessed using the MedCalc software package (Ver. 9.55, MedCalc Software, Mariakerke, Belgium) and R statistical software (Ver. 4.0.2). A univariate Cox proportional hazards model was evaluated to assess hazard ratios for selected potential predictors of OS. In order to adjust for the effects of other significant univariate factors, multivariate Cox proportional hazards models were applied to evaluate the independent factors for OS. The radiomics signature was applied as a LASSO score to establish a prediction model for OS. Kaplan-Meier analysis was used to compare the survival curves. 

We used the time-dependent receiver operating characteristic (ROC) curve in order to investigate the predictive performance of the LASSO score and compare it with conventional PET parameters [9,10]. The integrated area under the curve (IAUC) from time-dependent ROC curves was measured, and a larger IAUC means that the average predictability of time to event is higher. We also used decision curve analysis (DCA) to evaluate the net benefits of a range of threshold probabilities in clinical practice. In this study, several packages provided by the open-source statistical software R (http://www.R-project.org (accessed on 10 March 2021) were used as follows: “glmnet”, “time ROC”, “survminer”, “rmda”, and “ggDCA” packages. A two-sided *p* < 0.05 was considered significant. 

## 3. Results

### 3.1. Patient Characteristics

A total of 300 consecutive patients with stage III NSCLC were enrolled in this study (Figure 1). Patient characteristics are given in Table 1. The mean age was 60.0 years (range, 31–77), and most patients were male (*n* = 202, 67.3%). There was a high prevalence of adenocarcinoma (*n* = 211, 70.3%). The tumor stage was IIIA in 222 (74.0%), IIIB in 74 (24.7%), and IIIC in 4 patients (1.3%). The mean follow-up duration was 43.2 months, with a range of 3.2 to 150.9 months. At the time of analysis, 84 patients (28.0%) had died, and the remaining 216 patients (72.0%) were alive.

### 3.2. Conventional PET Parameters of Primary Tumors and Overall Survival

A univariate cox regression analysis of conventional PET parameters used to predict OS is listed in Table 2. The volume-based PET parameters, such as MTV and TLG, of both PET1 and PET2, were significant prognostic factors for OS; however, single-voxel parameters, such as SUVmax and SUVmean, failed to demonstrate prognostic significance for OS. In addition, the differences of all conventional PET parameters before and after neoadjuvant CCRT also did not show significant correlations with OS. Of the clinical characteristics, sex (male vs. female), histological cell type (non-adenocarcinoma vs. adenocarcinoma), T stage (T3/T4 vs. T1/T2), and tumor stage (IIIA vs. IIIB/IIIC) were all significant prognostic factors.

### 3.3. Feature Selection and LASSO Score for Predicting OS

The LASSO regression analyses showed that 14 radiomic features of PET1 and 18 radiomic features of PET2 were the most useful for predicting OS (Table 2). Each LASSO score for PET1 and PET2 was calculated based on those radiomic features. In univariate Cox regression analysis, both the PET1-LASSO score and the PET2-LASSO score were significant prognostic factors for OS in patients with stage III NSCLC (Table 3). 

Cut-off values for LASSO scores were assessed by ROC curve analysis. The optimal cut-off value for the PET1-LASSO score was −0.884 (*p* < 0.001; 95% confidence interval [CI], 0.607–0.717; AUC, 0.664) and was −0.737 for the PET2-LASSO score (*p* < 0.001; 95% CI, 0.618–0.727; AUC, 0.674). Kaplan-Meier curves for OS are shown in Figure 2. The survival curves showed that patients with higher LASSO scores had significantly poorer OS than those of the opposite group (both *p* < 0.001). 

### 3.4. Multivariate Survival Analysis

All statistically significant variables assessed by univariate analysis were indicated for multivariate Cox regression analysis. The results of a simple correlation test showed high multicollinearity between MTV and TLG, representing high variance inflation factors. Considering this result, multivariate analysis was performed by separating two models and selecting for MTV and TLG, respectively. In multivariate analysis, only the PET2-LASSO score was a significant independent prognostic factor after adjusting for sex, histological cell type, T stage, and tumor stage (Table 4). 

### 3.5. Assessment of Predictive Performance Using Time-Dependent ROC and Decision Curve Analysis

In order to compare predictive performance between volume-based PET parameters and LASSO scores, we used time-dependent ROC analysis. This method revealed the IAUC for each follow-up time (Table 5). The PET2-LASSO score revealed higher IAUCs than the PET1-LASSO score, as well as the volume-based PET parameters, in predicting the risk of death (*p* < 0.001), which demonstrated consistently better OS prediction performance during the follow-up period (Figure 3).

The DCA for the LASSO score model and the conventional PET parameters model are presented in Figure 4. DCA showed that the LASSO score model had a higher overall net benefit than the conventional PET parameter model across most of the risk threshold. Representative case with high LASSO score and poor prognosis is illustrated in Figure 5. 

## 4. Discussion

To the best of our knowledge, this is the largest cohort study to date evaluating prognostic factors in this patient group, which comprised of 300 patients. The estimation of OS in patients with stage III NSCLC is very important for determining a precise therapeutic strategy since the survival rate after treatment for stage III NSCLC still remains poor. We demonstrated that radiomic features using ^18^F-FDG PET/CT could be robust and useful in assessing the survival rate. The LASSO score calculated by radiomics provided prognostic significance for OS and predicted the survival rate more accurately than the conventional PET parameter model. We also observed that the radiomic signature of ^18^F-FDG PET/CT after neoadjuvant CCRT was an independent prognostic factor for NSCLC OS. 

Single-voxel values, such as SUVmax and SUVmean, are commonly used as semi-quantitative parameters for the assessment of tumor metabolic activity in clinical practice [21]. These parameters did not have significant prognostic value for NSCLC, and our findings are compatible with those of prior studies [22,23]. A previous study suggested that preoperative volume-based PET parameters, such as MTV and TLG, are significant prognostic factors for survival in stage III NSCLC [10]. In consistent with this result, univariate analysis in this study also revealed that MTV and TLG were statistically correlated with OS, as the patients with higher MTV and TLG had poorer OS than those with lower values of the parameters. However, these volume-based PET parameters did not show statistical significance, and only the LASSO score was an independent predictor for OS in multivariate analyses. It is presumed that the differences between them are probably because neither MTV nor TLG necessarily represents tumor biology. The reason for our finding is that radiomics can estimate detailed information on tumor heterogeneity and microenvironment, such as metabolic rate, hypoxia, necrosis, aggressiveness, and tumor cell proliferation, better than conventional volume-based PET parameters [24,25,26,27]. 

As we mentioned, the radiomic features provided information about the intratumoral heterogeneous characteristics by analyzing the quantitative variables. Since high-dimensional feature data with a quite large number of candidate predictors are evaluated, it is necessary to select relevant features to establish radiomics [28]. In this study, LASSO regression was used to select the radiomic features in order to minimize the influence of overfitting, as previously studied [29,30,31,32]. Our results are in agreement with those of earlier studies reporting that higher LASSO scores can be associated with poorer survival rates and distinguish patients into low- and high-risk groups with significant differences in survival. 

In order to apply the LASSO score using radiomics for the predictive performance of OS, we established time-dependent ROC analysis and DCA in clinical practice. We found that the prognostic performance of both LASSO scores using pre- and post-neoadjuvant treatment PET/CT was superior to that of the volume-based PET parameters regarding IAUC. The LASSO score models showed higher capabilities for OS risk assessment after curative surgery for stage III NSCLC and successfully classified those patients into high- and low-risk groups. The DCA also revealed that the LASSO score model was preferable to the conventional PET parameter model across most of the range of reliable threshold probabilities, suggesting that the radiomics signature enhanced the clinical significance for individualized OS estimation in patients with stage III NSCLC. 

The primary strength of our study is the analysis of radiomic features for survival prognosis in NSCLC, although several previous studies reported that texture analysis-extracted PET/CT had prognostic significance in many other cancers [12,20,27,31,32]. Moreover, this study demonstrated the important prognostic implications of radiomics in ^18^F-FDG PET/CT performed after neoadjuvant CCRT as well as pre-treatment examination. Radiomics is an emerging digital method for describing the cancer information that is associated with tumor size, shape, intensity, and texture and suggesting a more detailed complementary prognostic value compared with conventional PET parameters and clinicopathologic variables. Therefore, this current study might emphasize the prognostic significance of the radiomic signature in patients with stage III NSCLC. 

This study has some limitations. First, this study was performed in a retrospective design from a single center, and the results might be affected by selection bias. Therefore, we plan to acquire prospective multi-center data to investigate more general findings in the future. Second, because of the long retrospective study period, the patients were initially staged according to the AJCC’s 7th and 8th editions. Although there are some differences in the evaluation of T and M stages between these editions, we thought they were negligible due to there being no significant difference in overall stage III prevalence. Third, the possibility of pulmonary side effects originating from radiotherapy complicates radiomics measurement. In order to minimize this limitation, we tried our best to exclude post-radiotherapy changes such as pneumonitis or fibrosis, considering the relative intensity and the distribution of FDG uptake in lung parenchyma [33]. It is presumed that our acquired radiomics are regarded as consistent since the PET Edge technique has good reliability in radiomic analysis [34]. Fourth, a validation study was not performed with an independent dataset in this study, which might show the overestimated results in our prognostic model. Therefore, further validation using an independent dataset is also required to prove the generalizability of our results. 

## 5. Conclusions

In conclusion, the ^18^F-FDG PET/CT radiomic features of the primary tumor comprise an independent prognostic factor for the estimation of OS in patients with stage III NSCLC undergoing neoadjuvant CCRT followed by surgery. Furthermore, the newly developed LASSO score using radiomic features performs better for individualized OS estimation than conventional PET parameters. Radiomics using ^18^F-FDG PET/CT can be applied for the management of stage III NSCLC. These encouraging imaging biomarkers need standardization and validation through further multi-center prospective studies with larger patient cohorts.

## Figures and Tables

**Figure 1 cancers-15-02012-f001:**
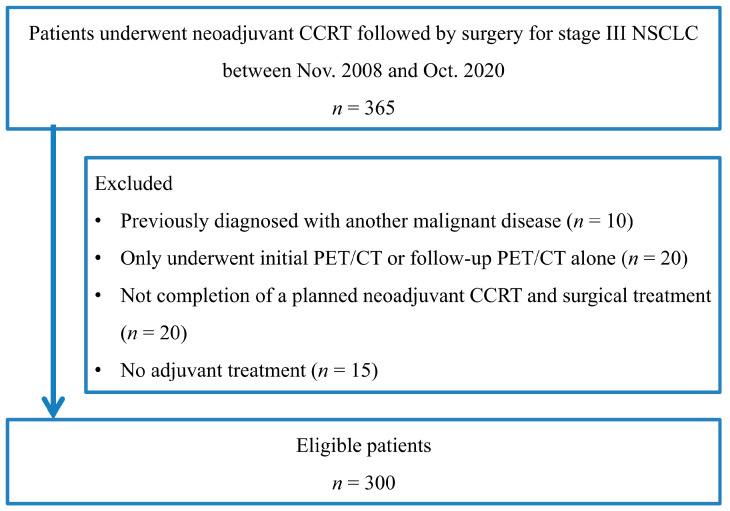
Flowchart shows patient selection.

**Figure 2 cancers-15-02012-f002:**
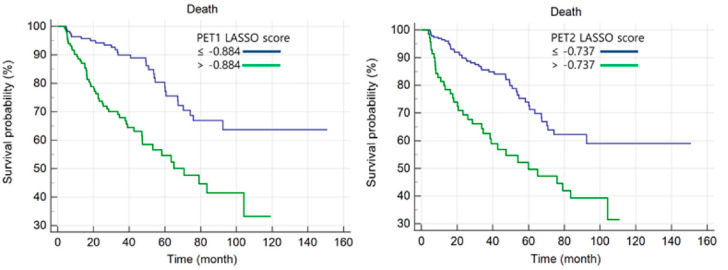
Kaplan-Meier analyses of PET1 and PET2 LASSO scores.

**Figure 3 cancers-15-02012-f003:**
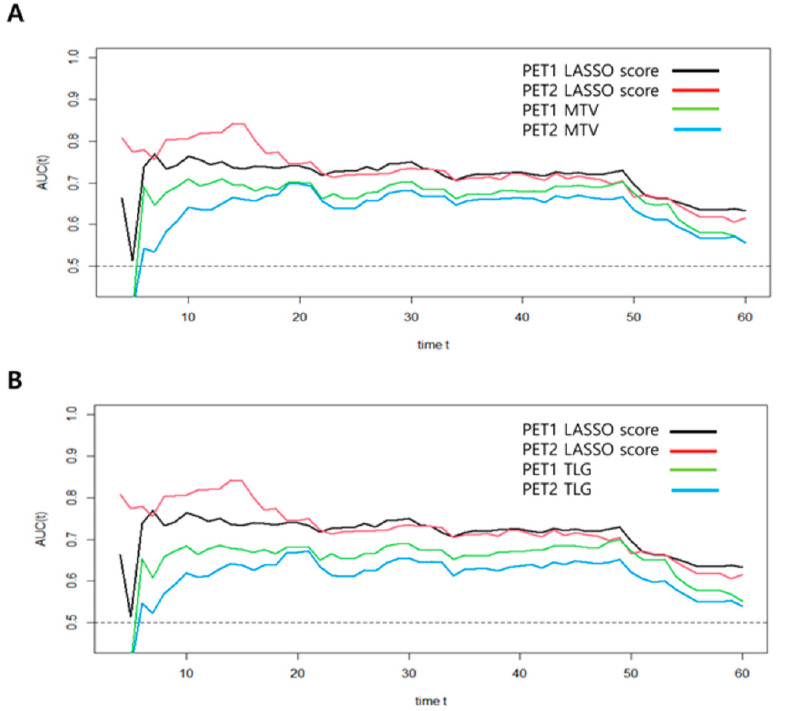
Time-dependent ROC curve analyses for the prediction of OS in patients with stage III NSCLC. (**A**) MTV model; (**B**) TLG model.

**Figure 4 cancers-15-02012-f004:**
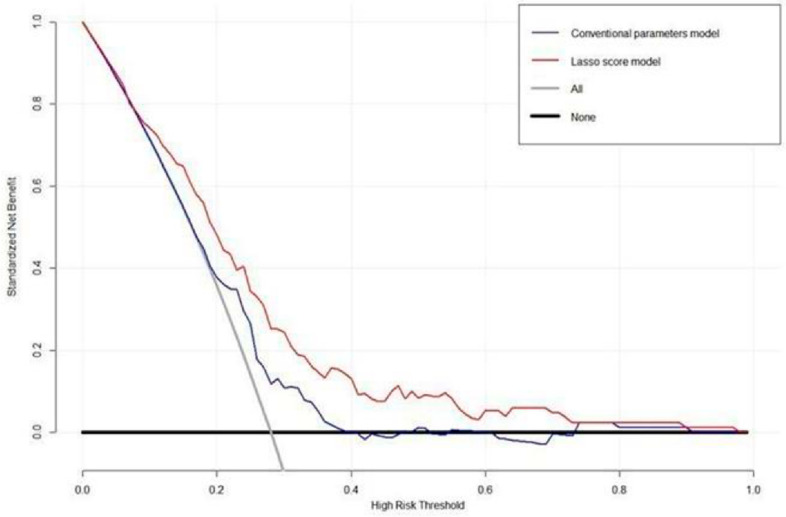
Decision curve analysis for LASSO score and a conventional PET parameter prediction model evaluated on patients with stage III NSCLC. The y-axis indicates the net benefit calculated by summing the benefits (true-positive findings) and subtracting the harms (false-positive findings). The blue line is the net benefit of predicting OS according to the conventional PET parameters, and the red line is the net benefit of predicting OS based on the LASSO score. DCA revealed that the prediction model for the LASSO score provided a superior net benefit compared with both the conventional PET parameter model and simple strategies, such as follow-up of all patients (gray line) or no patients (horizontal black line).

**Figure 5 cancers-15-02012-f005:**
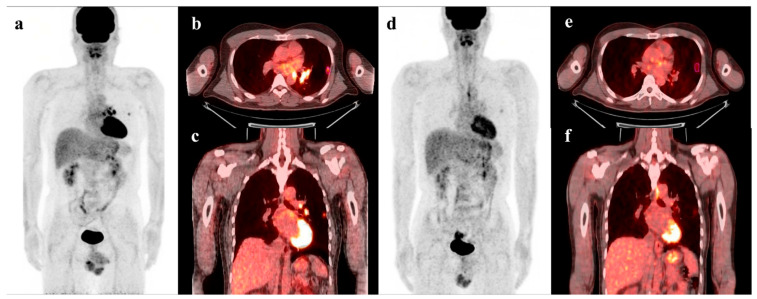
PET/CT images of a 44-year-old male patient with stage IIIa NSCLC show 1.2 cm-sized hypermetabolic tumor in the left lung upper lobe (pink segmented VOI, MTV = 1.20, TLG = 2.46, LASSO score = −0.496), and metastatic lymphadenopathy in left pulmonary hilar area and left mediastinum (**a**–**c**). Follow-up PET/CT images after neoadjuvant CCRT show increase in MTV and TLG, and decrease in LASSO score (MTV, 2.53; TLG 3.54; LASSO score −0.647) (**d**–**f**). This patient underwent left upper lobectomy with mediastinal lymph node dissection and adjuvant chemoradiotherapy. However, he died after 21.2 months of surgery.

**Table 1 cancers-15-02012-t001:** Patient characteristics.

Characteristic		Patients (%)
Age, mean (range), year		60.0 (31–77)
Sex	Male	202 (67.3)
	Female	98 (32.7)
Histology	Adenocarcinoma	211 (70.3)
	Squamous cell carcinoma	86 (28.7)
	Others	3 (1.0)
T stage	T1	89 (29.7)
	T2	134 (44.7)
	T3	57 (19.0)
	T4	20 (6.6)
N stage	N0	2 (0.7)
	N1	2 (0.7)
	N2	289 (96.3)
	N3	7 (2.3)
Tumor stage	IIIA	222 (74.0)
	IIIB	74 (24.7)
	IIIC	4 (1.3)
Type of surgery	Lobectomy	249 (83.0)
	Bilobectomy	15 (5.0)
	Pneumonectomy	11 (3.7)
	Lobectomy with en bloc wedge resection	25 (8.3)
Pathologic response	pCR	20 (6.7)
	Non-pCR	280 (93.3)
	MPR	144 (48.0)
	Non-MPR	156 (52.0)
PERCIST	CMR	44 (14.7)
	PMR	196 (65.3)
	SMD	59 (19.7)
	PMD	1 (0.3)
Adjuvant therapy	Chemotherapy	189 (63.0)
	Radiotherapy	92 (30.7)
	Chemoradiotherapy	19 (6.3)

pCR, pathologic complete response; MPR, major pathologic response; PERCIST, positron emission tomography response criteria in solid tumors; CMR, complete metabolic response; PMR, partial metabolic remission; SMD, stable metabolic disease; PMD, progressive metabolic disease.

**Table 2 cancers-15-02012-t002:** List of radiomic features using LASSO and LASSO score formula.

**PET1**	**Matrix**	**Index**
	Voxel-alignment matrix	Run percentage, low-intensity run emphasis
	Neighborhood intensity difference matrix	Busyness, strength
	Intensity size-zone matrix	Zone percentage, low-intensity zone emphasis, high-intensity short-zone emphasis
	Voxel statistics	SUV variance, SUV kurtosis, SUV kurtosis (bias corrected), tumor volume
	Texture spectrum	Max spectrum
	Texture feature coding co-occurrence matrix	Inverse difference moment, variance
**PET2**	**Matrix**	**Index**
	Co-occurrence matrix	Contrast
	Voxel-alignment matrix	Low-intensity long-run emphasis, high-intensity long-run emphasis
	Neighborhood intensity difference matrix	Complexity
	Intensity size-zone matrix	Zone percentage, low-intensity short-zone emphasis, low-intensity large-zone emphasis, high-intensity large-zone emphasis
	Voxel statistics	Minimum SUV, SUV skewness, SUV kurtosis, SUV skewness (bias corrected), entropy
	Texture spectrum	Max spectrum, black-white symmetry
	Texture feature coding	Coarseness
	Texture feature coding co-occurrence matrix	Second angular moment, intensity

PET1 LASSO score formula: 3.726224 − 0.04586424 × Run percentage − 2.355380 × Low intensity run emphasis − 0.3329271 × Busyness − 0.00922952 × Strength − 0.9670345 × Zone percentage − 4.623556 × Low intensity zone emphasis + 0.00001098790 × High intensity zone emphasis − 0.003274871 × SUV variance − 0.1388920 × SUV kurtosis + 0.1704623 × SUV kurtosis (bias corrected) + 0.004685532 × Tumor volume + 11.15391 × Max spectrum − 0.1288085 × Variance − 12.29401 × Inverse difference moment; PET2 LASSO score formula: 0.03060214 − 0.000002704935 × Contrast + 3.624602 × Low intensity long run emphasis + 0.0001780639 × High intensity long run emphasis − 0.004300564 × Complexity − 0.5597137 × Zone percentage − 4.516040 × Low intensity short zone emphasis + 0.02110051 × Low intensity large zone emphasis + 0.00009910182 × High intensity large zone emphasis + 0.4210314 × Minimum SUV − 0.02961810 × SUV skewness − 0.2418752 × SUV kurtosis − 0.0001231235 × SUV skewness (bias corrected) + 0.1595824 × Entropy + 0.07027946 × Black white symmetry + 6.237490 × Coarseness − 11.32393 × Second angular moment − 0.005009539 × Intensity.

**Table 3 cancers-15-02012-t003:** Univariate cox regression analysis for overall survival.

Variable		HR	95% CI	*p* Value
Age		1.016	0.991–1.043	0.216
Sex	Male vs. female	1.988	1.191–3.316	0.009 *
Histology	Non-ADC vs. ADC	1.595	1.023–2.487	0.039 *
T stage	T3/4 vs. T1/2	1.966	1.256–3.079	0.003 *
N stage	N2/N3 vs. N0/N1	1.110	0.154–8.006	0.917
Tumor stage	IIIb/IIIc vs. IIIa	2.067	1.329–3.215	0.001 *
Pathologic response	Non-pCR vs. pCR	1.741	0.839–3.615	0.137
	Non-MPR vs. MPR	1.045	0.680–1.605	0.841
PERCIST	SMD/PMD vs. CMR/PMR	1.193	0.692–2.058	0.525
PET1	SUVmax	1.015	0.975–1.056	0.476
	SUVmean	1.023	0.928–1.128	0.653
	MTV	1.005	1.002–1.008	<0.001 *
	TLG	1.002	1.001–1.003	0.005 *
	LASSO score	3.164	2.048–4.887	<0.001 *
PET2	SUVmax	1.008	0.941–1.080	0.823
	SUVmean	0.988	0.847–1.153	0.880
	MTV	1.010	1.003–1.016	0.003 *
	TLG	1.003	1.001–1.005	0.036 *
	LASSO score	2.836	2.102–3.826	<0.001 *
	%ΔSUVmax	1.003	0.994–1.011	0.538
	%ΔSUVmean	1.003	0.995–1.012	0.732
	%ΔMTV	1.001	0.998–1.002	0.876
	%ΔTLG	0.999	0.994–1.006	0.982

HR, hazard ratio; CI, confidence interval; ADC, adenocarcinoma; pCR, pathologic complete response; MPR, major pathologic response; PERCIST, positron emission tomography response criteria in solid tumors; CMR, complete metabolic response; PMR, partial metabolic remission; SMD, stable metabolic disease; PMD, progressive metabolic disease; *, *p* < 0.05.

**Table 4 cancers-15-02012-t004:** Multivariate cox regression analysis.

	HR	95% CI	*p* Value
MTV model			
Sex (male vs. female)	1.703	0.977–2.967	0.061
Histology (non-ADC vs. ADC)	1.309	0.793–2.162	0.293
T stage (T3/T4 vs. T1/T2)	1.589	0.217–1.822	0.629
Tumor stage (IIIb/IIIc vs. IIIa)	1.848	0.686–4.980	0.225
PET1 MTV (>32.23 vs. ≤32.23)	1.001	0.995–1.008	0.712
PET2 MTV (> 8.78 vs. ≤8.78)	0.994	0.981–1.008	0.393
PET1 LASSO score (>−0.884 vs. ≤−0.884)	1.707	0.907–3.212	0.097
PET2 LASSO score (>−0.737 vs. ≤−0.737)	2.297	1.437–3.669	<0.001 *
TLG model			
Sex (male vs. female)	1.674	0.960–2.919	0.067
Histology (non-ADC vs. ADC)	1.352	0.812–2.249	0.246
T stage (T3/T4 vs. T1/T2)	1.565	0.222–1.844	0.408
Tumor stage (IIIb/IIIc vs. IIIa)	1.863	0.694–5.005	0.217
PET1 TLG (>247.73 vs. ≤247.73)	0.999	0.999–1.001	0.883
PET2 TLG (>10.36 vs. ≤10.36)	0.999	0.994–1.004	0.670
PET1 LASSO score (>−0.884 vs. ≤−0.884)	1.787	0.950–3.362	0.072
PET2 LASSO score (>−0.737 vs. ≤−0.737)	2.084	1.419–3.060	<0.001 *

HR, hazard ratio; CI, confidence interval; *, *p* < 0.05.

**Table 5 cancers-15-02012-t005:** The IAUCs of volume-based PET parameters and LASSO score.

	12 Months	24 Months	36 Months	48 Months	60 Months
PET1					
MTV	0.598	0.646	0.654	0.658	0.653
TLG	0.573	0.625	0.635	0.639	0.637
LASSO score	0.695	0.715	0.718	0.719	0.707
PET2					
MTV	0.596	0.644	0.652	0.655	0.653
TLG	0.504	0.575	0.588	0.596	0.597
LASSO score	0.790	0.778	0.764	0.755	0.733

## Data Availability

Restrictions apply to the availability of these data. Data were obtained from the Samsung Medical Center and were available from the corresponding author with the permission of the Samsung Medical Center.

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
