# Peer review of "Radiomics Analysis of 18F-FDG PET/CT for Prognosis Prediction in Patients with Stage III Non-Small Cell Lung Cancer Undergoing Neoadjuvant Chemoradiation Therapy Followed by Surgery"

_cancers, 2023, doi:10.3390/cancers15072012_

Round 1

Reviewer 1 Report

The current study aims to estimate OS in patients with stage III NSCLC for therapy decision making. For this, a uni-centered database of 300 patients PET-CT images and their clinical data is used. They demonstrate that radiomic features using 18F-FDG PET/CT in the multi-variate model with LASSO score provided prognostic significance for OS and predicted survival rate more accurately than the conventional PET parameter model. Moreover, authoer use lontitudinal images : at T0 before the treatment and T1 3 weeks after the treatment.

The suggested method compares the conventional imaging paramereters (SUV, MTV, TLG), clinical parameters and multi-variate radiomics LASSO prediction score.

The subject is of interest to the clinical community, however, it is lacking few precisions. The prediction is performed on all the database available, it means that no independent tes was performed? Unfortunately, it has been shown that this overestimates the results. Your database is sufficient enough to perform a cross-validation, which would be in greater accordance with nowadays machien elarning approaches.

I would also think that it is interesting to compare clinical, imaging and radiomics models either all three in a univariate manner, or all three in a multi-variate framework.

It is also important to know what radiomics parameters were selected and their contribution score in the final multivariate LASSO.

Otherwise, the paper is quite clear and well structured.

Author Response

Dear editor of Cancers

We greatly appreciate the review of the original article and the helpful suggestions. Please, find below our point-by-point response to the reviewers’ comments and a decision of the changes made to the manuscript. Revisions in response to reviewer comments are shown in red in the revised manuscript.

Sincerely,

Joon Young Choi, M.D., Ph.D.

Department of Nuclear Medicine, Samsung Medical Center, Sungkyunkwan University School of Medicine, Seoul, Korea

Responses to Reviewer #1

Comments and Suggestions for Authors

The current study aims to estimate OS in patients with stage III NSCLC for therapy decision making. For this, a uni-centered database of 300 patients PET-CT images and their clinical data is used. They demonstrate that radiomic features using 18F-FDG PET/CT in the multi-variate model with LASSO score provided prognostic significance for OS and predicted survival rate more accurately than the conventional PET parameter model. Moreover, authoer use lontitudinal images : at T0 before the treatment and T1 3 weeks after the treatment.

The suggested method compares the conventional imaging paramereters (SUV, MTV, TLG), clinical parameters and multi-variate radiomics LASSO prediction score.

The subject is of interest to the clinical community, however, it is lacking few precisions. The prediction is performed on all the database available, it means that no independent test was performed? Unfortunately, it has been shown that this overestimates the results. Your database is sufficient enough to perform a cross-validation, which would be in greater accordance with nowadays machine learning approaches.

  • I understand your concern and agree with your idea. As you pointed out, we did not evaluate the prognostic performance by splitting the entire data into training/testing and independent sets. In obtaining the LASSO score, however, a 10-fold cross-validation was used instead of splitting the dataset. I will describe more details about this in the manuscript. In addition, I will study the prognostic performance of the independent set with multicenter prospective data in the future.

I would also think that it is interesting to compare clinical, imaging and radiomics models either all three in a univariate manner, or all three in a multi-variate framework.

  • Thank you for your great compliment.

It is also important to know what radiomics parameters were selected and their contribution score in the final multivariate LASSO.

  • As you said, it is essential to confirm what radiomics parameters were selected and how the LASSO scores were obtained using the selected parameters. We have listed them in detail in “Supplemental table 2”. Please refer to “Supplemental table 2”.

Otherwise, the paper is quite clear and well structured.

  • Thank you for your great compliment.

Reviewer 2 Report

This is exciting research with certain novelty in its content, reasonable research methods, reasonable description of the results, comprehensive and sufficient discussion, and concise language organization of the paper. However, some things could be improved in the interpretation of technical words in the article, and I still have some concerns about the article that needs to be addressed by the authors.

1.     The authors got the wrong definition of PMR. The correct definition is partial metabolic remission rather than pathological metabolic remission. (lines 190-191 and 208)

2.     We all know that the prognosis of tumors is related to the degree of pathological differentiation of tumors and whether high-risk patients choose postoperative adjuvant therapy. Therefore, the feasibility evaluation of any prediction model or parameters should be recommended to conduct correlation analysis with the existing accurate predictors. In other words, It is suggested that this paper further clarify whether the LASSO score positively correlates with the degree of pathological differentiation of patients and postoperative adjuvant therapy. However, since all patients in this paper have received postoperative adjuvant treatment, the supplementary postoperative adjuvant therapy does not make sense. Therefore, only relevant content related to the degree of pathological differentiation can be provided.

Author Response

Dear editor of Cancers

We greatly appreciate the review of the original article and the helpful suggestions. Please, find below our point-by-point response to the reviewers’ comments and a decision of the changes made to the manuscript. Revisions in response to reviewer comments are shown in red in the revised manuscript.

Sincerely,

Joon Young Choi, M.D., Ph.D.

Department of Nuclear Medicine, Samsung Medical Center, Sungkyunkwan University School of Medicine, Seoul, Korea

Responses to Reviewer #2

Comments and Suggestions for Authors

This is exciting research with certain novelty in its content, reasonable research methods, reasonable description of the results, comprehensive and sufficient discussion, and concise language organization of the paper. However, some things could be improved in the interpretation of technical words in the article, and I still have some concerns about the article that needs to be addressed by the authors.

  1. The authors got the wrong definition of PMR. The correct definition is partial metabolic remission rather than pathological metabolic remission. (lines 190-191 and 208)

à You are right. I typed incorrectly. I’ll fix it right.

  1. We all know that the prognosis of tumors is related to the degree of pathological differentiation of tumors and whether high-risk patients choose postoperative adjuvant therapy. Therefore, the feasibility evaluation of any prediction model or parameters should be recommended to conduct correlation analysis with the existing accurate predictors. In other words, It is suggested that this paper further clarify whether the LASSO score positively correlates with the degree of pathological differentiation of patients and postoperative adjuvant therapy. However, since all patients in this paper have received postoperative adjuvant treatment, the supplementary postoperative adjuvant therapy does not make sense. Therefore, only relevant content related to the degree of pathological differentiation can be provided.

à We agreed with you that the tumor differentiations is alleged prognostic factor. According to your comment, the tumor differentiations such as well-, moderate-, and poorly-differentiation were reviewed again. Among 300 patients, 274 were able to confirm the tumor differentiation, and the remaining 26 patients lacked tumor differentiation results or had limitations for evaluating the cell differentiation due to pathologic complete remission (pCR) after neoadjuvant CCRT. Of these 274 patients, only 3 cases showed well-differentiation. Moderate-differentiation was reported in 159 patients, and poorly-differentiation was in the remaining 112 patients. First, overall survival analysis was performed according to the tumor differentiation (Table). The attached table shows that tumor differentiation failed to demonstrate prognostic significance for OS. Also, the analysis of the relationship between tumor differentiation and LASSO score showed that there was a significant difference in PET1 LASSO score (p = 0.028) according to the tumor differentiation, but no significant difference in PET2 LASSO score (p = 0.991). This may be resulted from impossible evaluation for tumor differentiation in patients with pCR, which is well-known to be associated with good prognosis. Therefore, it does not seem to be appropriate to add these results in our study.

Variable

HR

95% CI

p value

Tumor differentiation

WD, MD vs. PD

0.913

0.577-1.44

0.698

Tumor differentiation

p value

WD, MD (n = 162)

PD (n = 112)

PET1 LASSO score (median, IQR)

-0.982 (-1.393 ~ -0.666)

-0.879 (-1.267 ~ -0.579)

0.029

PET2 LASSO score (median, IQR)

-1.092 (-1.550 ~ -0.749)

-1.130 (-1.540 ~ -0.754)

0.991

Thank you for your helpful comments.

Round 2

Reviewer 1 Report

Dear Authors,

Thank you for taking in consideration my previous comments and giving few mores details about the technical aspects of your method.

However, I think the final LASSO coefficient formula should be be brought to the main text, as it is the core of your method.

Moreover, even though you have performed the 10-fold cross validation, the fact that you haven't had your method tested on the independent data, overestimates greately your performance. This is very important to mention in your conclusion, for the future readers to take this point in consideration when comparing to your work.

Moreover, the English presents few mistakes : like line 213 : the most, line 214: was calculated. Unfortunately, I can't re-read fully, but you should improve the text by good English speaking full re-read.

Author Response

Dear editor of Cancers

We greatly appreciate the review of the original article and the helpful suggestions. Please, find below our point-by-point response to the reviewers’ comments and a decision of the changes made to the manuscript. Revisions in response to reviewer comments are shown in red in the revised manuscript.

Sincerely,

Joon Young Choi, M.D., Ph.D.

Department of Nuclear Medicine, Samsung Medical Center, Sungkyunkwan University School of Medicine, Seoul, Korea

Responses to Reviewer #1

Dear Authors,

Thank you for taking in consideration my previous comments and giving few mores details about the technical aspects of your method.

However, I think the final LASSO coefficient formula should be be brought to the main text, as it is the core of your method.

  • I understand your concern and agree with idea. The contents described in Supplementary Table 2 were re-written and inserted into the main manuscript as Table 2. As a result, the original table numbers are increased by one.

Moreover, even though you have performed the 10-fold cross validation, the fact that you haven't had your method tested on the independent data, overestimates greately your performance. This is very important to mention in your conclusion, for the future readers to take this point in consideration when comparing to your work.

  • We agreed with your comment. Relevant lines as another limitation were added in the Discussion section.

Moreover, the English presents few mistakes : like line 213 : the most, line 214: was calculated. Unfortunately, I can't re-read fully, but you should improve the text by good English speaking full re-read.

  • I will re-examine the grammar thoroughly. Thank you very much.